# A contact-electro-catalysis process for producing reactive oxygen species by ball milling of triboelectric materials

Ziming Wang[1,2,7], Xuanli Dong[1,2,7], Xiao-Fen Li[3,7], Yawei Feng[1,4], Shunning Li [5], Wei Tang [1,2] ✉ & Zhong Lin Wang [1,2,6] ✉

Ball milling is a representative mechanochemical strategy that uses the mechanical agitation-induced effects, defects, or extreme conditions to activate substrates. Here, we demonstrate that ball grinding could bring about contact-electro-catalysis (CEC) by using inert and conventional triboelectric materials. Exemplified by a liquid-assisted-grinding setup involving polytetrafluoroethylene (PTFE), reactive oxygen species (ROS) are produced, despite PTFE being generally considered as catalytically inert. The formation of ROS occurs with various polymers, such as polydimethylsiloxane (PDMS) and polypropylene (PP), and the amount of generated ROS aligns well with the polymers' contact-electrification abilities. It is suggested that mechanical collision not only maximizes the overlap in electron wave functions across the interface, but also excites phonons that provide the energy for electron transition. We expect the utilization of triboelectric materials and their derived CEC could lead to a field of ball milling-assisted mechanochemistry using any universal triboelectric materials under mild conditions.

Mechanochemistry has emerged as a promising field in recent decades due to its uniqueness for triggering/promoting chemical reactions[1–4]. Among various mechanochemical strategies, ball milling stands out for its versatility, scalability, and high efficiency[5–7], and has been extensively applied in redox reactions[8,9], mechanical alloying[10,11], and materials synthesis[12,13]. The activation of target substrates in conventional ball milling process usually relies on force-induced effects[14,15], increase of defects[16,17], or local extreme conditions[18,19] under external mechanical agitations. As a consequence, hard materials such as metals and ceramics that can withstand high-energy impact during grinding are widely employed for ball milling[20,21]. In addition to these well-established mechanisms, we have noticed that the ball milling process could naturally bring about frequent collisions even at low

revolution speeds, which is capable of inducing expedited contact-electrification (CE) phenomena through the use of triboelectric materials. The CE effect is ubiquitously existed among various interfaces, and recent studies have proved that electrons are the dominant charge carriers of CE in a majority of cases[22–24]. Besides, a high-intensity electric field at the contact surface could facilitate the electron transfer between different substrates as well as to motivate the subsequent reactions[25–27]. Contact-electro-catalysis (CEC) has been proposed to describe a catalytic process that is promoted by CE-driven interfacial electron transfer[28–30]. Ultrasonication was employed to initiate CEC in previous studies[31]. Alternatively, ball milling could directly provide frequent contact-separation cycles and the mechanical energy could be straightforwardly conveyed to substrates without water bath and

[1]CAS Center for Excellence in Nanoscience, Beijing Institute of Nanoenergy and Nanosystems, Chinese Academy of Sciences, Beijing 100140, China. [2]School of Nanoscience and Engineering, University of Chinese Academy of Sciences, Beijing 100049, China. [3]Key Laboratory of Advanced Materials (MOE), School of Materials Science and Engineering, Tsinghua University, Beijing 100084, China. [4]Department of Mechanical Engineering, City University of Hong Kong, Hong Kong 999077, P. R. China. [5]School of Advanced Materials, Shenzhen Graduate School, Peking University, Shenzhen 518055, China. [6]School of Materials Science and Engineering, Georgia Institute of Technology, Atlanta, GA 30332-0245, USA. [7]These authors contributed equally: Ziming Wang, Xuanli Dong, Xiao-Fen Li. ✉e-mail: tangwei@binn.cas.cn; zhong.wang@mse.gatech.edu

related effects. Therefore, we anticipate the triboelectric materials and their derived CEC could lead to a field of ball milling-assisted mechanochemistry using universally available triboelectric materials.

Here, exemplified by a typical liquid-assisted grinding (LAG) process, we demonstrate that the presence of triboelectric materials can catalyze the generation of reactive oxygen species (ROS). To unambiguously investigate the contribution of CE, the LAG setup is made of pristine polymers, such as polytetrafluoroethylene (PTFE), polydimethylsiloxane (PDMS), or polypropylene (PP), which are rarely utilized as catalysts. We propose that the CE brought by frequent collisions during grinding can drive electron exchange between pristine polymers and surrounding substrates ($H_2O$, or $O_2$ for example), thereby catalyzing the formation of ROS. The generation of ROS is confirmed by the electron paramagnetic resonance (EPR) technique, and the resulting concentration varies according to the CE performance of employed materials. Specifically, PTFE that exhibits the highest CE performance shows the supreme performance, followed by PDMS. In contrast, the PP group, which can hardly be electrified, produces a negligible amount of ROS. The consistence in performances of these polymers in producing ROS and CE indicates the dominant role of CEC. Further investigations reveal that the grinding process not only offers frequent collisions for triggering CE, but also provides physical impact to facilitate the CE-driven electron transfer by increasing the overlap in electron wave functions and exciting phonons to provide the energy for electron transition. The reactivity of

CEC-yielded ROS is verified by degrading methyl orange (MO), wherein a 50 mL of 5-ppm MO aqueous solution is completely degraded in the PTFE group after grinding at 350 revolution per minutes (RPM) for 120 min, and CEC-produced ROS is identified as the primary reason for the degradation. We envision the employment of triboelectric materials can fully exploit the naturally occurred collisions in grinding, and it-derived contact-electro-catalysis may lower the energy input required for promoting reactions, forming an efficient and promising strategy for mechanochemistry. This is a rather universal process because triboelectric materials are very abundant in nature.

## Results

### Producing reactive oxygen species (ROS) via CEC in ball milling

Figure 1a presents a schematic illustration of employing triboelectric materials for mechanochemistry by ball milling. The overall architecture closely resembles a conventional liquid-assisted grinding (LAG) setup, except the vial and ball are made of triboelectric materials. We postulate frequent collisions that occur naturally during milling could induce evident contact-electrification (CE) phenomena, and the CE-driven interfacial electron transfer is supposed to promote the rate of chemical reactions, i.e., contact-electro-catalysis (CEC)[28]. The catalytic ability of CE effect was discussed in Fig. 1b by comparison with well-established piezoelectric effect. The piezoelectric effect has been extensively applied for a series of significant mechanochemical reactions, such as formation and regeneration of active Cu[I] species in

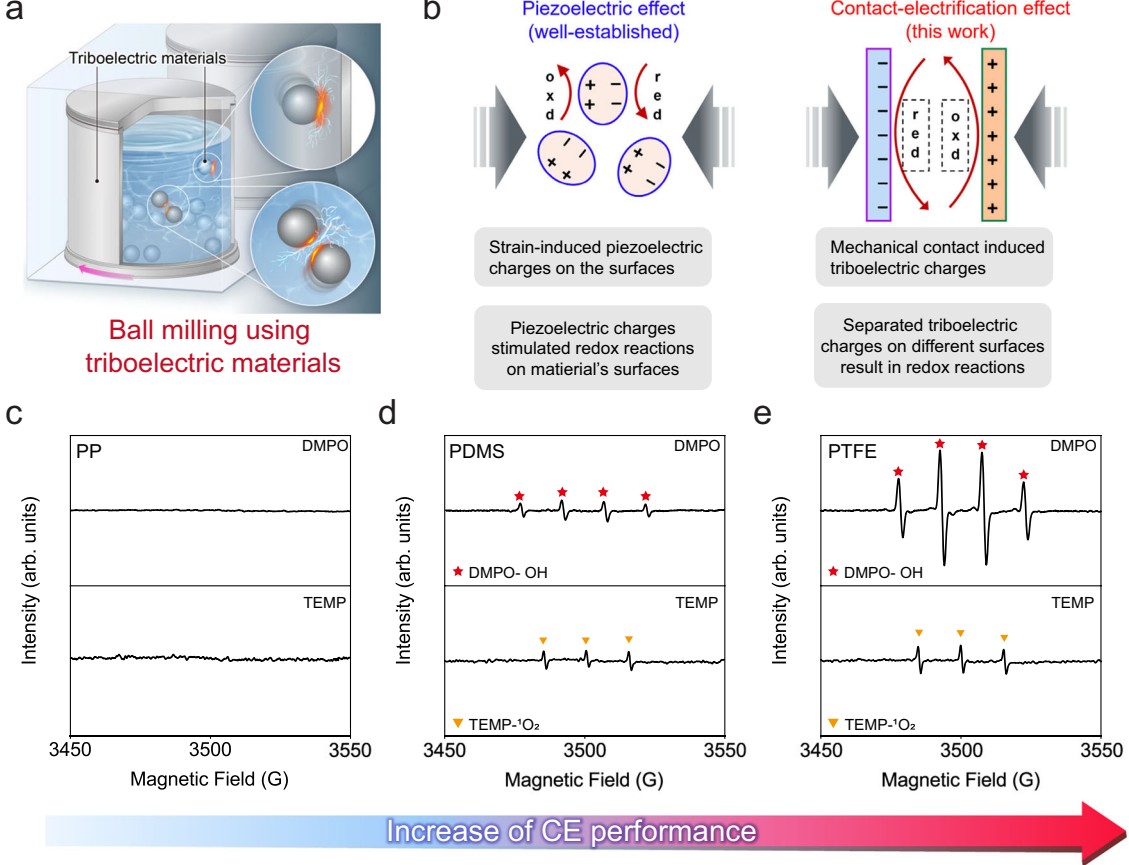

**Fig. 1 | Production of reactive oxygen species (ROS) via contact-electro-catalysis (CEC) in ball milling. a** Schematic illustration of a ball mill process using triboelectric materials. **b** Comparison between piezoelectric and contact-electrification (CE) effects, which suggests that catalyzing reactions by CE charges should be possible. The abbreviation "red" represents for reduction reactions, and "oxd" for oxidation reactions. **c**–**e** Measured electron paramagnetic resonance (EPR) spectra in the condition of employing different triboelectric materials for ball milling. PP is short for polypropylene, PDMS for polydimethylsiloxane, and PTFE for polytetrafluoroethylene. 5,5-dimethyl-1-pyrroline N-oxide (DMPO) is the capture for hydroxyl and superoxide radicals, and 2,2,6,6-tetramethyl-4-piperidone hydrochloride (TEMP) is employed for capturing singlet oxygen. Source data are provided as a Source Data file. Created with Procreate, PowerPoint, and Adobe Illustrator.

atom transfer radical cyclization[32], reversible addition-fragmentation chain transfer polymerization[33], and aryl-amination reactions[34]. The underlying mechanism is usually summarized as a single electron transfer (SET) process[35]. To be specific, the agitation of piezoelectric materials under external stress could generate highly polarized charges on the surface[8], and these piezoelectric charges could assist the separation and segregation of electrons from holes, resulting in subsequent redox reactions[36,37]. Similarly, due to the contact-electrification effect, charges are produced once two surfaces come into contact[38-40], and the quantity of triboelectric charges generally exceeds those produced by the piezoelectric effect[41]. Recent studies have revealed that electrons participate in, and in most cases, dominant the charge transfer mechanism of CE[23,24]. Besides, the charged surface because of CE would result in an electric field in space, and a high-intensity electric field at the contact interface could significantly affect ambient environments[25-27]. For example, the high-intensity electric field at the surface of water microdroplet is supposed to promote electron transfer for the formation of hydroxyl radicals and $H_2O_2$[42-44]. Such electric field has also been found at the water-oil contact interfaces, and it-driven electron transfer could catalyze the formation of reactive oxygen species (ROS) for effective degradation of hexadecane[27]. A higher reaction rate could be obtained by intensifying the electric field at the surface[44]. These experimental observations further support the feasibility of catalyzing chemical reactions by CE. Additionally, triboelectric materials are much more universal than piezoelectric materials[45,46], so that the applications of CEC are much broader.

As a proof-of-concept investigation, we have selected three representative triboelectric materials, namely polypropylene (PP), polydimethylsiloxane (PDMS), and polytetrafluoroethylene (PTFE) to construct the ball milling setup. Ultrapure water was used as the grinding media, and vials as well as balls were made of the same material in each group. (Supplementary Note 1) For example, in the PTFE group, both vials and balls were made of PTFE. To investigate the correlation between CE strength and mechanochemical reaction rates, we first quantified the CE performance of different materials by a single-electrode mode triboelectric nanogenerator (SE-TENG), as illustrated by Supplementary Fig. 1. Measured transferred charges in Supplementary Fig. 2 indicate that PTFE exhibits the highest CE ability upon contact with water, while PP can hardly be electrified, which can be attributed to the varying electron-withdrawing (EW) abilities based on functional groups of polymers[47]. The catalytic ROS generation performance of these three materials were characterized by the electron paramagnetic resonance (EPR) technique, and the results were demonstrated from Fig. 1c–e. PP, which exhibits the lowest CE ability, serves as the control group, and no obvious peak was detected. However, the characteristic quadruplet peak that belongs to hydroxyl radicals (·OH) was observed in both PDMS and PTFE groups, verifying the effective generation of ·OH. Further, the peak intensity of ·OH generated by PTFE group is higher than that by PDMS, which is consistent with measured difference in CE abilities. As shown in Supplementary Fig. 3, the sextuplet characteristic peak of superoxide radicals (·$O_2^-$) after being captured was not detected until 1 mM ter-butanol was introduced, which might be ascribed to that the existence of ter-butanol could quench some ·OH radicals and thus enhance the opportunity for ·$O_2^-$ radicals to be captured[48]. The production of ·$O_2^-$ radicals was confirmed by using TEMP to capture $^1O_2$, the oxidative products of ·$O_2^-$ radicals[49]. The triplet peak of TEMP-$^1O_2$ was found in both PDMS and PTFE groups, and the intensity in PTFE group also surpasses that in PDMS group, as demonstrated in the bottom panel in Fig. 1c–e. The generation of ROS, and the consistence between the concentration of ROS and CE performance of used materials not only suggests the feasibility of CEC in mechanochemistry, but also indicates the dominant role of CE performance in the production of ROS.

## Investigation on the reactivity of CEC-yield ROS

Exemplified by the degradation of methyl orange (MO), we have exploited the reactivity of CEC-produced ROS. 50 mL of 5-ppm MO aqueous solutions and 100 g PTFE balls were transferred together into a PTFE vial for grinding. The revolution speed of vial was set as 350 RPM, and 4 minutes of pause for every 6 minutes of milling to minimize the heat effect. Optical and corresponding thermographic images in Fig. 2a, b clearly showed a notable decolorization of MO after grinding for 120 mins, with the highest temperature recorded on the internal wall of vial being 23.7 °C. The thermal stability of MO depicted in Supplementary Fig. 4 indicates that thermal decomposition has a negligible effect on MO decolorization. To investigate the underlying mechanism, aliquot (1 mL) was sampled at specific milling intervals and subjected to UV-Vis spectroscopy as well as liquid-chromatography mass-spectroscopy (LC-MS) analysis. Figure 2c demonstrated the characteristic absorbance intensity of MO in UV-Vis spectra decreased as the milling time increased, and approached zero after 60 mins. The LC-MS results in Fig. 2d proved that the decolorization mainly results from chemical degradation. Further analysis on the mass spectra in Supplementary Fig. 5 indicates the MO has undergone an oxidative degradation process during milling. As illustrated by Fig. 2e, the evolution of MO relative concentration in presence of various scavengers revealed the contribution of both hydroxyl and superoxide radicals to the degradation of MO. Moreover, we have determined the size of PTFE balls before and after degradation using Image J, as detailed in Supplementary Fig. 6. Statistical analysis of the results presented in Fig. 2f implies that no discernible variation in shape deformation and size distribution was observed after grinding. The chemical stability of PTFE balls was verified through ex-situ spectroscopic analysis techniques (Supplementary Fig. 7). In virtue of the stability in both physical and chemical properties, we have investigated the recyclability of PTFE balls, and found that no obvious decay in the performance for the degradation has been observed after recycling PTFE balls for 5 times (Supplementary Fig. 8).

In addition to PTFE, we have also performed a series of parallel degradation experiments under identical conditions except the balls and vials are made of PP or PDMS. As demonstrated in Fig. 2g, PTFE that exhibits the highest CE ability is the best performer, followed by PDMS. However, only 3.1% of MO was removed in PP group that can hardly withdraw electrons from water. The consistence between the CE abilities and degradation rates suggests the reactivity of CEC-produced ROS. Besides, degradation by conventional ball mill setups was also investigated as shown in Supplementary Fig. 9a. $ZrO_2$ is supposed to provide a much higher energy impact than pristine polymers can do during grinding, while exhibiting an almost negligible CE ability in contact with water (Supplementary Fig. 9b). Around 12.04% degradation rate was achieved in in $ZrO_2$ group that should mainly be ascribed to the strong energy impact during ball milling. In contrast, the degradation rate in PDMS and PTFE groups could respectively reach 32.05% and 99.18%. This result not only implies that defects or extreme conditions brought by high-energy impact are not the primary reason for degradation in PDMS and PTFE groups, but also indicates the utilization of triboelectric materials and it-derived CEC may offer an efficient strategy for ball milling under mild conditions.

A schematic illustration was proposed in Fig. 2h to explicate the generation of ROS by triboelectric materials-enabled CEC and its application in oxidative degradation. Owing to the presence of triboelectric materials, frequent collisions during grinding are accompanied by obvious contact-electrification effect and it-derived electron exchange. For example, according to the triboelectric series, electrons will be transferred to PTFE once it contacts with $H_2O$. Such electron exchange will give rise to the formation of water radical cations that will convert to hydronium cations and hydroxyl radicals through a rapid proton transfer from water[50]. Electrons left on PTFE surface will be captured by dissolved $O_2$, forming superoxide radicals. Hydroxyl

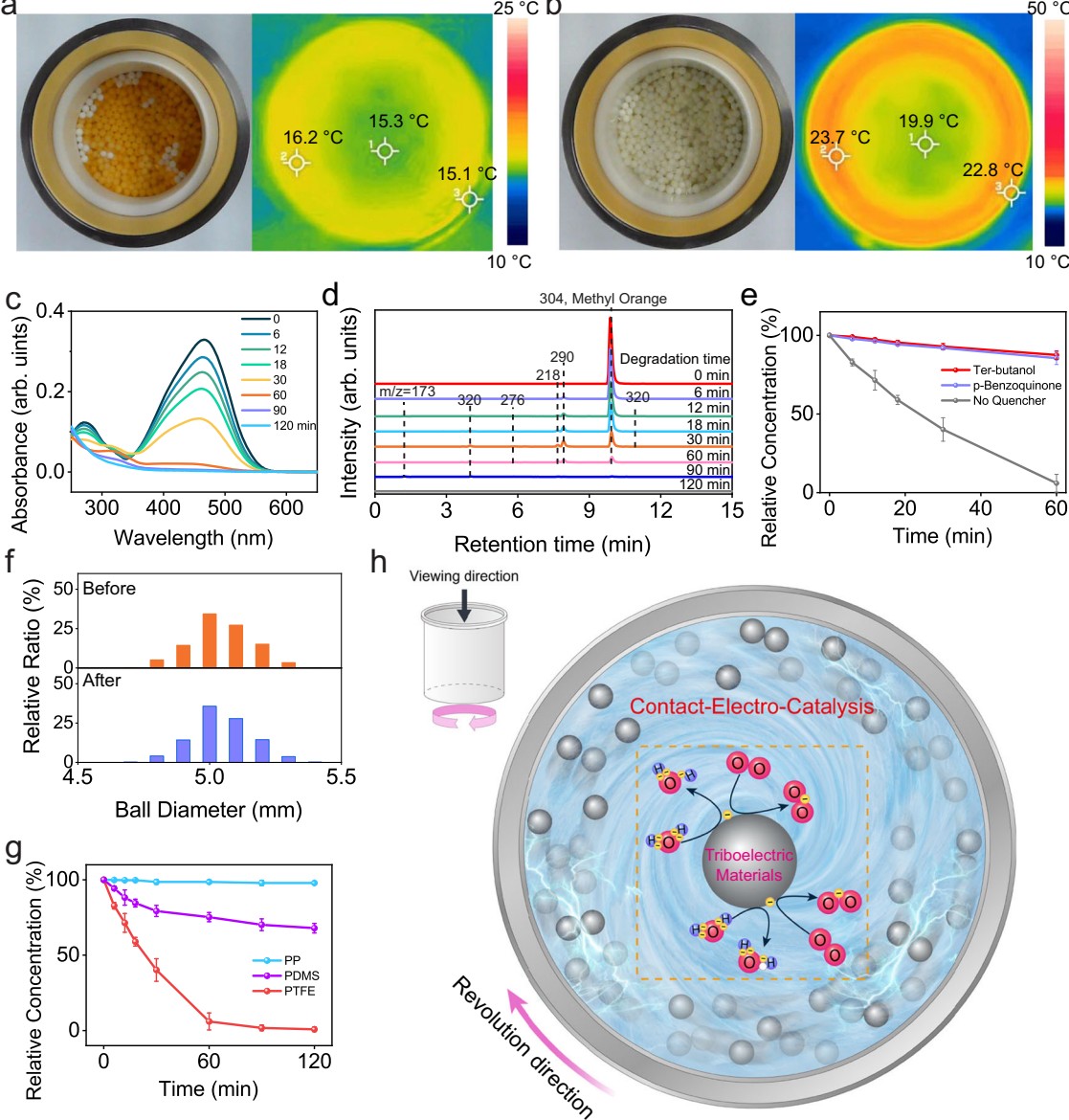

**Fig. 2 | Degrading methyl orange (MO) aqueous solution by contact-electro-catalysis (CEC)-yield reactive oxygen species (ROS).** Optical and thermographic images of polytetrafluoroethylene (PTFE) vials and balls (**a**) before reaction, as well as (**b**) after the reaction. **c** UV-Vis spectra of a 5-ppm MO aqueous solution when PTFE vial and balls were employed for ball milling. **d** Liquid chromatograph of the aqueous MO solution at different degradation time intervals with peaks being identified by mass spectra. **e** Evolution of relative concentration of MO under various radical scavengers with the same final concentration of 1 mM. The relative concentration is calculated based on the absorbance of MO that measured by UV-Vis spectroscopy. **f** Size distribution of PTFE milling balls before and after grinding.

**g** Performance comparison on MO decolorization in presence of different pristine polymers, including PTFE, polydimethylsiloxane (PDMS) and polypropylene (PP). The relative concentration is calculated based on the absorbance of MO that measured by UV-Vis spectroscopy. **h** Proposed working principle of CEC during ball milling using triboelectric materials. The gray circle represents for milling balls made of triboelectric materials, pink circle for O atoms, purple circle for H atoms, and yellow circle for electrons. Error bars represent standard deviation based on three replicate data. Source data are provided as a Source Data file. Created with Procreate and Adobe Illustrator.

and superoxide radicals generated at the end of both steps then react with organic pollutants in aqueous solution.

## Evolution of CEC efficiency under various revolution speeds

The revolution speed is a crucial parameter in ball milling that significantly affects the impact mode/energy among balls and vials, and thus leads to distinct catalytic efficiencies. Taking the degradation of MO as a model reaction, we have investigated the correlation between revolution speed and CEC efficiency. Figure 3a presents the evolution of relative concentration of MO in PTFE group under various revolution speeds. MO can barely be degraded when the revolution speed

is below 100 RPM. However, a 19.5 times higher decline of MO relative concentration is found once the revolution speed arises to 150 RPM, and the degradation rate increases further with higher revolution speeds. Theoretical analysis on kinetics of ball milling indicates the variation in revolution speed affects both the collision frequency and impact energy. Details can be found in Supplementary Note 2. Nearly no degradation was detected in condition of 50 RPM even the grinding time was elongated to 600 mins (Supplementary Fig. 10), suggesting that the low collision frequency is not the dominant reason for poor catalytic efficiency at low revolution speeds.

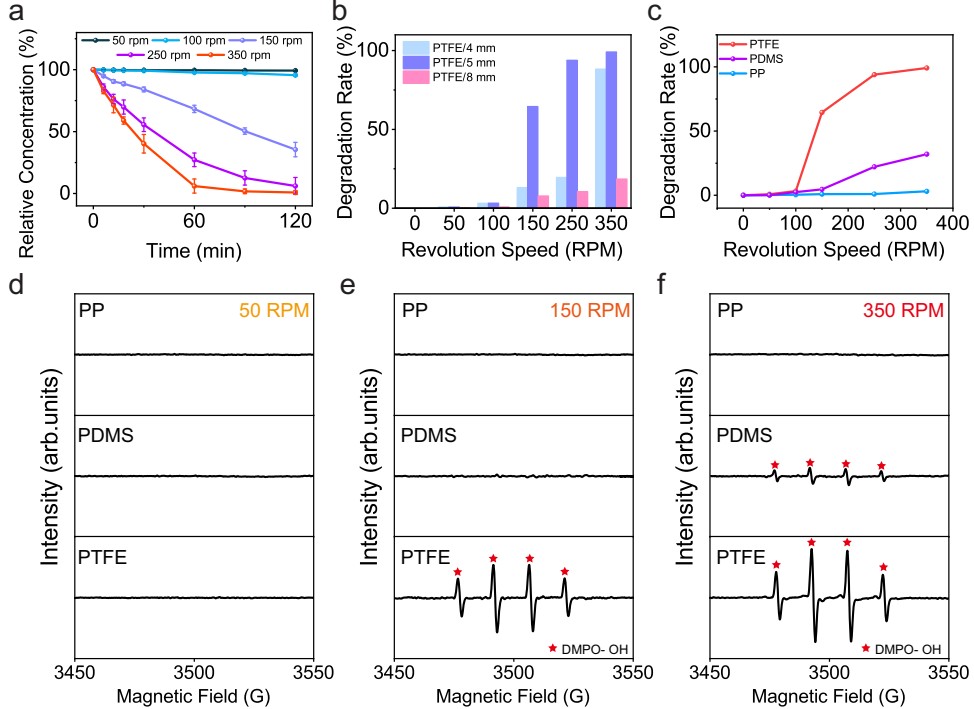

**Fig. 3 | Investigations on the relationship between milling speed and degradation rate. a** Evolution of relative concentration of methyl orange (MO) aqueous solution in condition of different milling speeds. The relative concentration is calculated based on the absorbance of MO that measured by UV-Vis spectroscopy. **b** Influence of ball sizes on the degradation rate. PTFE is short for polytetrafluoroethylene, and the subsequent number is the diameter of used PTFE balls. The degradation rate is defined as the percentage of reduced absorbance of MO. **c** Comparison of degradation rate when different triboelectric materials were utilized. PP is short for polypropylene, and PDMS for polydimethylsiloxane. The size of milling balls is 5 mm in this investigation. **d–f** EPR profiles of different groups under 50 RPM (**d**), 150 RPM (**e**), and 350 RPM (**f**) revolution speeds. Error bars represent standard deviation based on three replicate data. Source data are provided as a Source Data file.

The contribution from impact energy was investigated by using PTFE balls with different sizes, as depicted in Fig. 3b. The total weight of PTFE balls was given as 100 g. The 5-mm group exhibits a faster degradation than 4-mm group, which might be ascribed to the higher impact energy brought by 5-mm group. Although an even higher impact energy could be obtained by further increasing the size of balls to 8 mm, the measured degradation rate decays significantly, which might result from a drastic decrease of CE frequency. The outperformance of 5-mm group indicates there exists an optimal diameter to reach a balance between impact frequency and energy, as specified in Supplementary Note 3. Besides, the evolution of degradation rate under various revolution speeds for PDMS and PP groups was also examined as illustrated in Fig. 3c. Obvious degradation was achieved in PTFE group when the revolution speed was beyond 150 RPM. In comparison, no obvious degradation was detected until the revolution speed reaches 250 RPM in PDMS group. Although very limited degradation performance was presented in PP group under all given revolution speeds, its degradation rate also increases as the revolution speed arises (Supplementary Fig. 11). Clearly can we find that a material exhibiting higher CE performance could simultaneously obtain a higher degradation rate and a lower speed threshold for initiating CEC. To verify this assumption, EPR was employed to characterize the generation of ROS under various combinations of materials and revolution speeds. No peak was found in PP, PDMS, and PTFE groups when the revolution speed was set as 50 RPM, as shown in Fig. 3d. Waveforms in Fig. 3e show that the production of ROS was found only in the PTFE group when the revolution speed was 150 RPM, a speed that was beyond the threshold for PTFE group but below that required for PP and PDMS groups. Figure 3f depicts that as the revolution speed increases to 350 RPM, although still no obvious peak is detected in PP group, both PTFE and PDMS groups can produce ROS, and the

intensity in PTFE group is apparently higher than that in PDMS. Such results not only prove that a material with higher CE ability could facilitate the generation of ROS through CEC, but also indicate the CEC may be initiated at even lower speeds by improving the CE ability of used materials, providing an alternative strategy for building mechanochemical setups toward milder conditions.

## Analysis on enhanced CEC efficiency under elevated revolution speeds

An electron cloud-potential-well model has been introduced to elucidate the catalytic enhancement achieved under elevated revolution speeds. When two media are brought into contact through an external force, local high pressure is generated at the points of contact, even at the atomic scale[38]. As the applied force increases, the distance between two atoms belonging to different media decreases, leading to an increased overlap of their electron clouds, as depicted in Fig. 4a. This stronger electron cloud overlap is supposed to reduce energy barriers for electron exchange. Consequently, more electrons will be transferred when the CE process is subjected to higher contact forces. The validity of this assumption was first experimentally verified by measuring the transferred charge quantities under various contact forces. To precisely control the contact force, we have developed a contact-separation mode TENG and then attached it to a linear motor that could perform cyclic contact with preset contact force. The profiles obtained from Fig. 4b clearly show that a higher contact force gives rise to an increased quantity of electrons being transferred during CE, which could then facilitate the induction and exchange of electrons during CEC.

Furthermore, we have simulated the electron redistribution of PTFE chains and $H_2O/O_2$ molecules at different distances to verify the above assumption. Since larger forces are indicative of closer

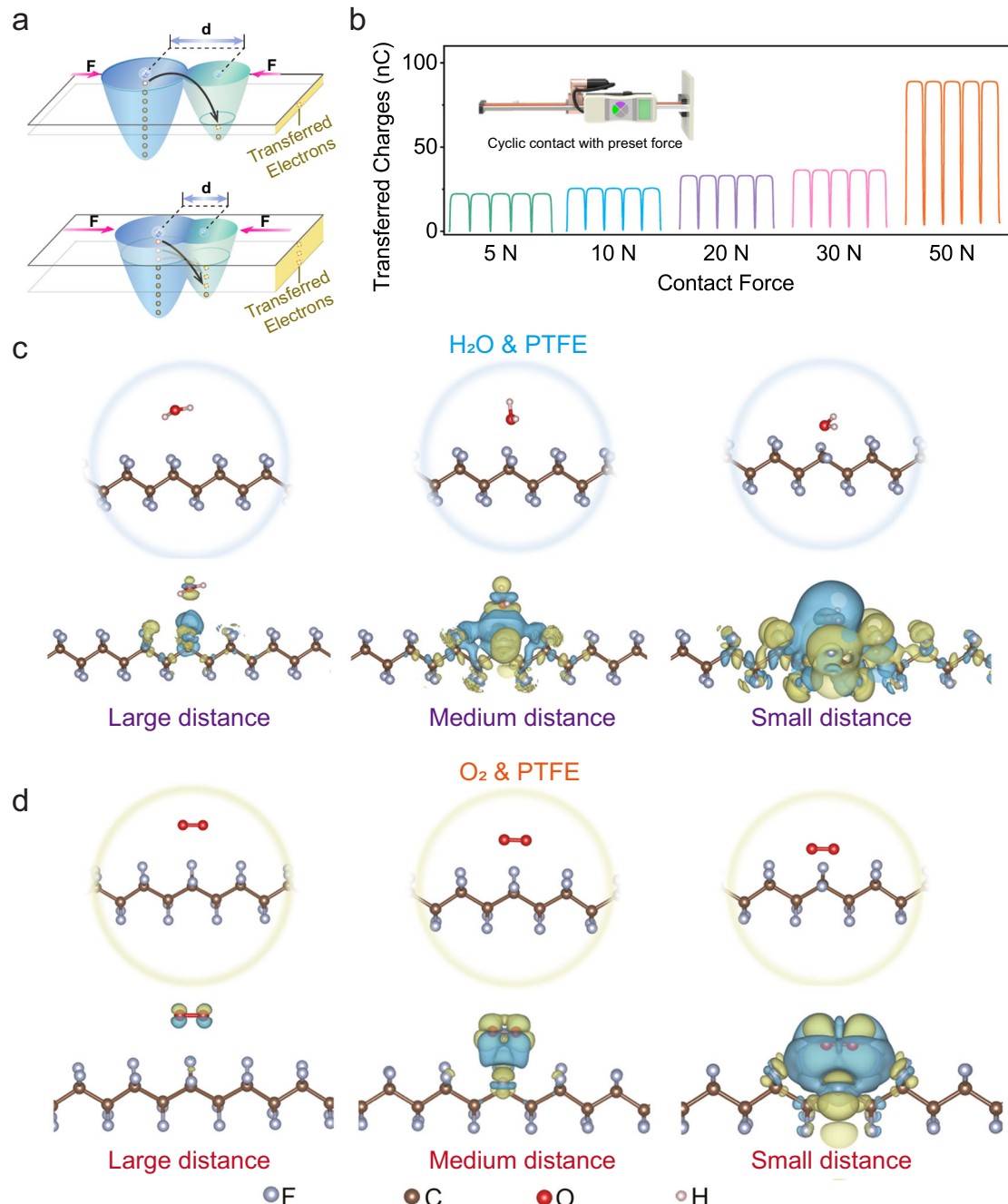

**Fig. 4 | Investigations on contact force and electron transfer process during contact-electrification. a** Schematic view of electron transfer during contact-electrification at atomic scale under different contact forces. The electron clouds of two atoms are depicted in light blue and green colors, respectively. More electrons (denoted by yellow circle) will be transferred once the distance (d) between these two atoms diminishes as the result of higher applied force (F). **b** Measured transferred charges when a polytetrafluoroethylene (PTFE) film repeatedly contacts with a Cu film under various given contact forces. The inset depicts a linear motor that links with a force sensor for performing cyclic contact under various given forces. **c** Simulated spatial distribution of charge density difference between $H_2O$ molecules and PTFE chains at different distances. **d** Simulated spatial distribution of charge density difference between $O_2$ molecules and PTFE chains at different distances. Yellow and blue isosurfaces denote areas of charge accumulation and depletion ($\pm0.0002\,e\,Å^{-3}$), respectively. Figure 4c, d was produced with VESTA[51]. Source data are provided as a Source Data file. Created with SolidWorks and Adobe Illustrator.

distances, the smaller distance condition is regarded as an equivalent to stronger contact force. Detailed insights into the simulation can be found in Supplementary Note 4. Figure 4c depicts the simulated spatial distribution of the charge density difference between $H_2O$ molecules and PTFE chains at varying distances respectively. No significant charge redistribution is observed when PTFE chains were separated from $H_2O$ molecules, indicating a negligible interaction between them. However, the extent of charge accumulation and depletion expands along with the decrease of distance between PTFE chains and $H_2O$ molecules, and a more pronounced charge redistribution was observed as the separating distance was further reduced, signifying more robust interactions were developed between these two entities. A similar trend was observed in the PTFE/$O_2$ model as shown in Fig. 4d, which also presents an increased interaction as the distance between PTFE and $O_2$ molecules decreased. Based on above theoretical analysis, we postulate that the proximity of different molecules could affect the

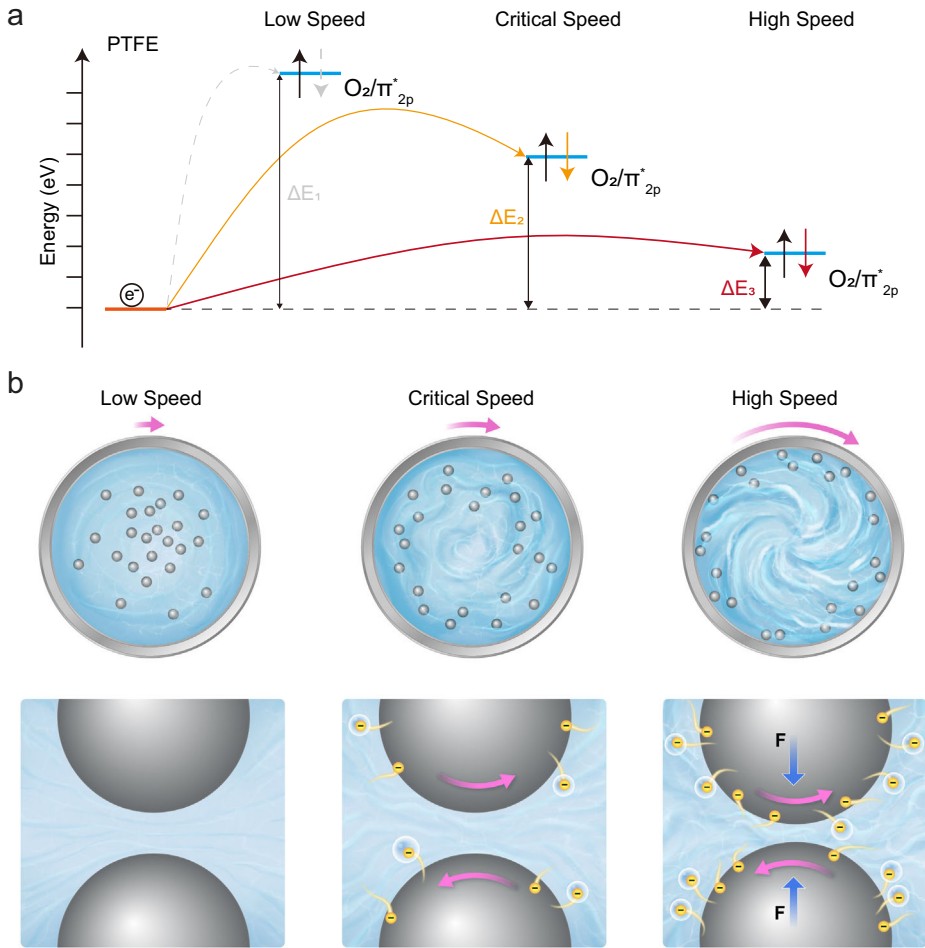

**Fig. 5 | Mechanistic analysis of contact-electro-catalysis. a** Schematic energy diagram for electron exchanging between polytetrafluoroethylene (PTFE) chains and $O_2$ molecules. The orange line represents the energy level of electrons (indicated by $e^-$ in this figure) of PTFE, while the blue line refers to the energy level of antibonding π orbit of $O_2$. Arrows denotes the process of transferring electrons from PTFE to $O_2$. **b** Proposed schematic for elucidating the enhanced catalytic performance at elevated revolution speed. The top panel exhibits a top view of the grinding setup under various revolution speeds, and the bottom section demonstrates corresponding status of milling balls. "F" refers to force that drives two milling balls getting closer at high revolution speeds. Negative charges in yellow circles represent for electrons, the blue background for the liquid environment, and the pink arrow for revolution direction of milling vials or balls. Created with Procreate and Adobe Illustrator.

strength of interactions and thus electron transfer process between them. As a consequence, more electrons will be exchanged during CE as a result of the higher contact force/impact energy brought by elevated revolution speeds. Besides, the collision frequency during ball milling also rises with the enhancement of revolution speeds, as illustrated in Supplementary Note 2. Improvements in both sides could contribute to an enhanced CEC efficiency at elevated revolution speeds.

**Study on the threshold for initiating CEC**

Another intriguing phenomenon we have noticed during ball milling is that there might exist a speed threshold for initiating the catalytic process. For example, the degradation of MO is not obvious until the revolution speed beyond 100 RPM in PTFE group. Taking the electron transfer procedure between PTFE chains and $O_2$ molecules as an example, we have simulated the total density of state (TDOS) of electron donor (PTFE) and partial density of state (PDOS) of electron acceptor ($O_2$) in a co-existence model with different distances to elucidate this variation. Similarly, the higher impact energy at enhanced revolution speeds is assumed to induce a closer distance between PTFE chains and $O_2$ molecules. The molecular orbital configurations of $O_2$ molecules and superoxide radicals in Supplementary Fig. 12a, b

indicate that the electron from PTFE should be transferred to the antibonding π orbit of $O_2$ for producing superoxide radicals. The energy level of each orbit under five different separating distances was calculated and depicted together in Supplementary Fig. 12c. It clearly can we find that a lower energy position of antibonding π orbit was obtained as the distance decreased, and Supplementary Fig. 12d confirmed that the energy barrier for such electron transition was decreased from 3.8839 eV to 2.5384 eV, implying that less energy was required for generating superoxide radicals. Figure 5a summarizes the correlation between revolution speeds and corresponding electron transition process. At low revolution speeds, the low impact energy is unlikely to overcome the high-energy barrier ($\Delta E_1$) to drive electron exchange for producing superoxide radicals, thereby presenting a negligible degradation rate of MO. As the revolution speed increased, not only the impact energy was improved, but also the energy barrier for electron transition was declined. Thus, we speculate that there exists a threshold speed at which the energy is sufficient for electrons to just overcome the energy barrier for yielding superoxide radicals ($\Delta E_2$). This critical speed is referred to the speed threshold for initiating CEC. Such energy difference continues to diminish ($\Delta E_3$) as the revolution speed further increases, resulting in a more significant formation of superoxide radicals for MO degradation. According to the

qualitive model established in Supplementary Note 2, the specific values of impact energy of 5-mm PTFE balls under various revolution speeds were calculated and listed in Supplementary Table 1.

A schematic illustration was proposed in Fig. 5b to explicate the presence of speed threshold and the evolution of CEC efficiency under various speeds. The small frequency of contact-separation cycle and impact energy at low revolution speeds can hardly drive electron exchange for producing ROS. However, the formation of ROS is feasible once the revolution speed beyond a threshold value. Further increase in revolution speeds not only provides more opportunities for contact-electrification between milling balls and surrounding substrates by elevated collision frequencies, but also enables more electrons to be exchanged during CE by high impact energy. Benefitted from enhanced CE frequency and the quantity of exchanged electrons, a more significant CE-driven interfacial electron transfer was attained with increased revolution speeds, leading to an enhanced catalytic efficiency. Mechanical collision can also generate phonons in solid materials. The existence of a threshold speed may indicate that a hard collision can not only maximize the interatomic electron wave-function overlap across the interface, but also produce high-energy phonons that provide the energy required for interfacial electron transition.

## Discussion

In virtue of frequent collisions that naturally occurred during ball milling, a triboelectric material-based grinding setup was first proposed for mechanochemical catalysis. The utilization of triboelectric materials could induce contact-electrification (CE) phenomena, thereby promoting reaction rates through contact-electro-catalysis (CEC). Our results demonstrated that CE-driven electron exchange during grinding could produce ROS, and the concentration of generated ROS aligned well with the measured difference in CE abilities. Further investigations revealed that the milling process not only offers frequent collisions to trigger CE, but also provides impact energies for facilitating CE-driven electron transfer by maximizing the interatomic electron wave-function overlap and reducing the interfacial energy barriers. Thus, an enhanced catalytic efficiency is achieved at elevated revolution speeds, and a threshold speed exists for initiating CEC. Furthermore, triboelectric materials are very popular and broad that include almost all non-metallic inorganic materials and organic materials, so the process we have elaborated here is rather universal. The employment of triboelectric materials and the resulting contact-electro-catalysis enable a broader range of materials that could be utilized to construct mechanochemical setups and enrich the category of catalytic mechanisms that could be applied in mechanochemical catalysis, so that mechanochemistry can be much more expanded.

## Methods

### Instrumentation and chemicals

Materials were obtained from commercial suppliers and were not subject to further modifications unless otherwise noted. Methyl orange [$C_{14}H_{14}N_3NaO_3S$, Macklin, 98%], p-benzoquinone [$C_6H_4O_2$, Macklin 99.5%], Isopropanol [$C_3H_8O$, Macklin, 99.9%], polytetrafluoroethylene [Dupont, $(C_2F_4)n$, 4 mm, 5 mm, 8 mm, ≥99.9%], polydimethylsiloxane [Dupont, $(C_2H_6OSi)n$, 5 mm, 99.9%], polypropylene [Dupont, $(C_3H_6)n$, 5 mm, 99.9%], and Zirconium (IV) oxide [Fritsch, $(ZrO_2)$, 5 mm, 99.9%], 5,5-Dimethyl-1-pyrroline N-Oxide [Dojindo, $C_6H_{11}NO$], and 2,2,6,6-Tetramethyl-4-piperidone hydrochloride [Dojindo, $C_9H_{17}NO.HCl$] were purchased from the corresponding suppliers. All reactions were conducted using corresponding grinding vessels in a Planetary Mill PULVERISETTE 5 premium line (Fritsch Inc. Germany).

### Sample preparation

A 5-ppm aqueous solution of methyl orange was prepared by adding 5 mg of methyl orange ($C_{14}H_{14}N_3NaO_3S$) to 1 L of ultrapure water, followed by magnetic stirring for 1 hour.

100 g milling balls were added together with 50 mL of an as-prepared methyl orange aqueous solution into corresponding vials. The duty ratio of the planetary mill was set as 60%.

Aliquots were sampled at 0, 6, 12, 18, 30, 60, 90, 120 mins intervals during grinding for the subsequent UV-Vis measurements.

Samples for EPR analysis were prepared by applying 20 ml of ultrapure water and 40 g of balls in a ball milling process. About 0.25 ml of DMPO was transferred to the solution prior to ball milling.

The balls after reactions were separated from solution using a filtration system. The filtered balls were first washed by ultrapure water and then dried in an oven at 40 degrees overnight before analysis.

FTIR samples were prepared by grinding 10 mg of polymer balls with 100 mg of KBr and then pressing them into a pellet.

### Sample characterization

The absorbance of the aliquot was recorded on a Cary 3500 UV-Visible Multicell, and the scanning wavelength range is 250–650 nm. The samples were put into a Hellma Analytics QS High precision cell with the light path of 10 mm.

Pictures of thermography was obtained by using FLIR One Pro thermographic camera (FLIR System Inc.)

The X-ray photoelectron spectroscopy measurements were conducted on a Thermo Fisher Scientific K-Alpha. The Alka-ray source (hv = 1486.6 eV) was used in a vacuum of $1 \times 10^{-9}$ mBar with an operating voltage of 15 kV and a filament current of 10 mA. The pass energy was set at 30 eV.

The Raman spectroscopy analysis was conducted on a LabRam HR evolution (HORIBA, SAS France), using a range from 300 to 1400 $cm^{-1}$.

FTIR analyses were conducted on a Bruker Vertex 80 v on a range from 400 to 3000 $cm^{-1}$.

The LC-MS analysis was conducted on using a Thermo Scientific Q Exactive Orbitrap Quadrupole-Electrostatic Field Orbitrap High Resolution Tandem Mass Spectrometer. The HESI ion source of the mass spectrometer was set at −3.0 kV, in positive ion mode. The mass spectrometry scanner was set on full scan range 100–1000 m/z. The resolution of the instrument is 70000 FMHM. The column used was a Hypersil Gold C18 ($2.1 \times 100$ mm, 1.9 μm), the column temperature is set at 40 °C. The injection volume is 5 μL. The mobile phase A is composed of 0.1% formic acid aqueous solution, and the mobile phase B is an acetonitrile solution.

Electron paramagnetic resonance was recorded on a Bruker EMX plus-9.5/12/P/L. The measurements were performed in X-band (9.827635 GHz) with amplitude modulation of 1 G, microwave power of 2 mW, amplitude modulation frequency of 100 kHz, conversion time of 60 ms, and time constant of 40.96 ms.

## Data availability

The data supporting the findings of this study are reported in the main text or the Supplementary Information. Source data are provided as a Source Data file. Source data are provided with this paper.

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

## Acknowledgements

We would like to thank Jiahui Zeng and Shuxing Xu for their productive discussions and assistance with collecting and organizing data. National Key R&D Project from Minister of Science and Technology (Grant no. 2016YFA0202704, Z.L.W.); National Natural Science Foundation of China (Grant Nos. 51432005, and 5151101243, Z.L.W.); Beijing Municipal Science and Technology Commission (Grant Nos. Z181100003818016, Z171100000317001, Z171100002017017, Y3993113DF, T.W.); Youth Innovation Promotion Association, Chinese Academy of Sciences (T.W.); China Postdoctoral Science Foundation (Grant No. 2022M723101, Z.W.).

## Author contributions

W.T. and Z.L.W. conceived the idea and supervised the experiment. Z.W., X.D., X.F.L., Y.F., W.T., and Z.L.W. prepared the manuscript. Z.W. and X.D. developed the experimental setups. Z.W. and X.D. performed data measurements. X.F.L. and S.L. performed theoretical simulations. All the authors discussed the results and commented on the manuscript.

## Competing interests

All the authors declare no competing interests.
