## [Peer Review File · Nature Communications]

A contact-electro-catalysis process for producing reactive oxygen species by ball milling of triboelectric materialsREVIEWER COMMENTS

Reviewer #1 (Remarks to the Author):

Tang, Wang, and their research colleagues have provided compelling evidence demonstrating the generation of reactive oxygen species through the utilization of triboelectric materials under mechanochemical conditions. Mechanochemistry has emerged as a highly promising field in recent decades, offering unique activation pathways that remain inaccessible through conventional means such as heating or light irradiation. In particular, the ball-milling technique has gained widespread popularity in synthetic chemistry due to its remarkable versatility, scalability, and impressive efficiency. Notably, some research groups have ventured into the realm of mechanoredox reactions, harnessing the power of piezoelectric materials in conjunction with ball milling to initiate chemical transformations. Simultaneously, the authors have been actively exploring the frontier of contact-electro-catalysis using triboelectric materials, a departure from their prior focus on ultrasonication. In this study, they have successfully extended their innovative approach to encompass ball milling reactions, marking a significant advancement in their research journey.

This manuscript is exceptionally well-written, and the experiments conducted are characterized by a systematic and meticulous approach. Furthermore, the mechanistic studies carried out in this work provide a comprehensive understanding of the underlying mechanisms. In light of these strengths, this reviewer wholeheartedly recommends the publication of this study, with the understanding that a few minor revisions are needed.

While the authors have certainly acknowledged the existence of redox reactions employing piezoelectric materials and ball milling, the references cited are limited in number. To enhance the scholarly context of the work, it is advisable to incorporate relevant literature, particularly the paper mentioned below:

DOI: 10.1002/anie.202003565

DOI: 10.1002/anie.202215733

DOI: 10.1002/anie.202311531

DOI: 10.1002/anie.202009844

Reviewer #2 (Remarks to the Author):

In this paper, the authors demonstrate that ball grinding could intrigue contact-electrocatalysis (CEC) by using inert and conventional triboelectric materials. The results revealed that the higher CEC properties can produce a higher generation of ROS. The work focused on the inert materials can produce the CEC performance in the presence of water, which is helpful to understand the generation of ROS in the advanced oxidation process. I think it would be interesting for the readers of catalyst. However, there are a few questions need to be addressed to improve the novelty and readability of manuscripts. The details are listed as below:

1. The author declared that a contact-electrification (CE) phenomena would be induced between the inert and conventional triboelectric materials during the friction process, However, there are also a few reports, such as, Science, 366 (2019) 1500; Nano Energy, 94 (2021) 106890, reported that when grinding the piezo materials and other chemicals the piezo catalytic process also can produce the ROS, so what is the biggest difference between the piezo catalytic and the triboelectric catalytic process. More introductions are necessary.

2. What is the structure difference between the triboelectric material and the piezo materials? How to define a material with triboelectric properties?

3. line 167: Most molecules can be decomposed or degradation under the strong energy impact, it is not an except for ZrO₂, so is the high degradation performance produced by the ZrO₂ due to the strong collision of the balls or the CE performance? Before the experimental, the author should consider whether ZrO₂ is a kind of triboelectric material?

4. Author declaredly that “PTFE exhibits the highest CE performance and show the supreme performance, followed by PDMS, the PP group can hardly be electrified, thus produced a negligible amount of ROS”. It is not strict, whether the H₂O or O₂ molecular can be reduced largely depend on the LUMO orbit of the material, so the reduction ability of PTFE, PDMS and PP should be considered.

5. line 206. By using PTFE balls with different sizes, the contribution from impact energy was investigated. The result shows that the higher size radius exhibits a faster degradation due to the higher impact energy, however, with increase on balls size leads

to a decay in degradation rate, and the author explained that a drastic decrease of CE frequency. So, what is the main contribution on a high degradation performance, a higher CE frequency or the stronger impact energy?

6. The correlation ship among the generation of ROS, the impact energy and the CEC performance need to deep considered. And further, Fig 5a, to make more clearly, the specific value of impact energy should be given.

Reviewer #3 (Remarks to the Author):

The results are noteworthy, in my opinion.

It does add to the existing literature in the field concerning contact electrification. I also find the methodology to be sound, but I suggest that the presentation could be improved by discussing some additional references that I think are relevant to this topic.

The spontaneous electrification of surfaces and interfaces is a widespread phenomenon that produces unexpected effects

in mass and charge transfer, phase change, and chemical reactivity, as revealed in abundant literature from the past twenty

years. I was disappointed to find no references to the Brazilian scientists Fernando Galembeck and Andre Galembeck. I also urge that consideration be given to the recent publication: Colussi, A. J., Mechanism of Hydrogen Peroxide Formation on Sprayed Water Microdroplets. *J. Am. Chem. Soc.*, 2023, 145, 16315–16317 which I think is relevant to how contact electrification works.

Response to Referees' Comments

Reviewer #1 (Remarks to the Author):

Comments: Tang, Wang, and their research colleagues have provided compelling evidence demonstrating the generation of reactive oxygen species through the utilization of triboelectric materials under mechanochemical conditions. Mechanochemistry has emerged as a highly promising field in recent decades, offering unique activation pathways that remain inaccessible through conventional means such as heating or light irradiation. In particular, the ball-milling technique has gained widespread popularity in synthetic chemistry due to its remarkable versatility, scalability, and impressive efficiency. Notably, some research groups have ventured into the realm of mechanoredox reactions, harnessing the power of piezoelectric materials in conjunction with ball milling to initiate chemical transformations. Simultaneously, the authors have been actively exploring the frontier of contact-electro-catalysis using triboelectric materials, a departure from their prior focus on ultrasonication. In this study, they have successfully extended their innovative approach to encompass ball milling reactions, marking a significant advancement in their research journey.

This manuscript is exceptionally well-written, and the experiments conducted are characterized by a systematic and meticulous approach. Furthermore, the mechanistic studies carried out in this work provide a comprehensive understanding of the underlying mechanisms. In light of these strengths, this reviewer wholeheartedly recommends the publication of this study, with the understanding that a few minor revisions are needed.

While the authors have certainly acknowledged the existence of redox reactions employing piezoelectric materials and ball milling, the references cited are limited in number. To enhance the scholarly context of the work, it is advisable to incorporate relevant literature, particularly the paper mentioned below:

DOI: 10.1002/anie.202003565

DOI: 10.1002/anie.202215733

DOI: 10.1002/anie.202311531

DOI: 10.1002/anie.202009844

Response: We sincerely appreciate the reviewer's precious time and effort in evaluating our manuscript. We fully agree with your constructive suggestion on including more significant pioneer research works in the manuscript, which would greatly enhance the depth and breadth of our literature review. Corresponding revisions have been made and highlighted by yellow background in the main text.

Once again, we would like to express our gratitude for your positive comments and kind support to strengthen our manuscript. We look forward to the opportunity to share our revised manuscript with you and the broader scientific community.

Revision 1: Following revisions have been made in the Introduction.

Mechanochemistry has emerged as a promising field in recent decades due to its uniqueness for triggering/promoting chemical reactions.¹⁻⁴

Revision 2: Following revisions have been made in the main text.

The catalytic ability of CE effect was discussed in Fig. 1b by comparison with well-established piezoelectric effect. The piezoelectric effect has been extensively applied for a series of significant mechanochemical reactions, such as formation and regeneration of active Cu^I species in atom transfer radical cyclization,³² reversible addition-fragmentation chain transfer polymerization,³³ and aryl-amination reactions.³⁴ The underlying mechanism is usually summarized as a single electron transfer (SET) process.³⁵ To be specific, the agitation of piezoelectric materials under external stress could generate highly polarized charges on the surface,⁸ and these piezoelectric charges could assist the separation and segregation of electrons from holes, resulting in subsequent redox reactions.^{36,37}

Reviewer #2 (Remarks to the Author):

Comments:

In this paper, the authors demonstrate that ball grinding could intrigue contact-electro-catalysis (CEC) by using inert and conventional triboelectric materials. The results revealed that the higher CEC properties can produce a higher generation of ROS. The work focused on the inert materials can produce the CEC performance in the presence of water, which is helpful to understand the generation of ROS in the advanced oxidation process. I think it would be interesting for the readers of catalyst. However, there are a few questions need to be addressed to improve the novelty and readability of manuscripts. The details are listed as below:

Response: We are very grateful for the reviewer's constructive suggestions that greatly help us to enhance the quality of our manuscript. We sincerely apologize for some of our texts and figures are not presented clearly enough in our previous submission. We have systematically revised them based on your insightful comments, and supplemented necessary information to enhance the readability. We are confident that the revised version has been much improved, and all your concerns have been properly addressed. The itemized responses and revisions are as follows:

General issues:

Comment 2.1: The author declared that a contact-electrification (CE) phenomena would be induced between the inert and conventional triboelectric materials during the friction process, However, there are also a few reports, such as, Science, 366 (2019) 1500; Nano Energy, 94 (2021) 106890, reported that when grinding the piezo materials and other chemicals the piezo catalytic process also can produce the ROS, so what is the biggest difference between the piezo catalytic and the triboelectric catalytic process. More introductions are necessary.

Response 2.1: Thanks so much for your valuable advice. We believe the biggest difference between piezocatalysis and contact-electro-catalysis is their catalytic mechanisms as well as it-derived selection range of catalysts. Piezoelectric materials such as BaTiO₃ and Cu₃B₂O₆ have been widely employed in pioneer research on piezo-driven mechanoredox reactions. Strain-

induced piezoelectric charges on the surface are responsible for stimulating subsequent redox reactions through a process known as single electron transfer (SET). Therefore, catalysts in piezocatalysis must be piezoelectric materials, which are specific groups of materials. In contrast, contact-electrification (CE) occurs for almost all of materials since it is a universal effect. Recent studies have proved that electron is the dominant charge carrier during CE, and the high-intensity electric field at contact interfaces could facilitate electron transfer between different substrates as well as to motivate subsequent reactions. Contact-electro-catalysis has been proposed as an emerging and effective strategy for mechanoredox reactions by utilizing the CE-driven electron transfer. Owing to the CE effect is ubiquitously existed among various interfaces (Mater. Today, 2019, 30: 34-51; Chem. Rev., 2021, 12: 5209-5232), the materials that could be envisaged as catalysts in CEC have been greatly expanded. For example, by utilizing the CE-driven electron transfer at the surface of PTFE, ROS could be produced even PTFE is chemically inert to a majority of chemicals and itself has rarely reported with any catalytic activity. Besides, as the CE effect is mainly a surface phenomenon that remains largely independent of the bulk structure/property of materials, the CE ability and corresponding CEC performance could be facilely improved by proper surface modifications. Recent investigations suggest that the CEC performance of SiO₂ and MOF (MIL-101 (Cr)) could be enhanced by introducing high CE-performance groups on their surfaces. (Nano Energy, 2023, 108198; Nano Energy, 2023, 108433) Thus, the contact-electro-catalysis is not only an important supplement to current mechanoredox strategies, but it also results in a much broad selection range of catalysts.

Revision 3: Following revisions have been made in the Introduction and main text.

The CE effect is ubiquitously existed among various interfaces, and recent studies have proved that electrons are the dominant charge carriers of CE in a majority of cases.²²⁻²⁴ Besides, a high-intensity electric field at the contact surface could facilitate the electron transfer between different substrates as well as to motivate the subsequent reactions.²⁵⁻²⁷ Contact-electro-catalysis (CEC) has been proposed to describe a catalytic process that is promoted by CE-driven interfacial electron-transfer.²⁸⁻³⁰ ... Therefore, we anticipate that the utilization of triboelectric materials and their derived CEC could lead to a field of ball milling-assisted mechanochemistry using universally available triboelectric materials.

Besides, the charged surface because of CE would result in an electric field in space, and a high-intensity electric field at the contact interface could significantly affect ambient environments.²⁵⁻²⁷ ... Furthermore, triboelectric materials are much more universal than piezoelectric materials,^{45,46} so that the applications of CEC are much broader.

Comment 2.2: What is the structure difference between the triboelectric material and the piezo materials? How to define a material with triboelectric properties?

Response 2.2: Thanks so much for your precious suggestion. The piezoelectric effect is a bulk property that originates from the displacement of the positive and negative ions (instead of electrons) within the material under external mechanical stimuli. Therefore, the piezoelectric materials are usually characterized with a non-centrosymmetric crystal structure, such as BaTiO₃, ZnO, and etc. In contrast, the triboelectric/ contact-electrification effect is a ubiquitous

surface phenomenon that remains largely independent of the bulk structure/property of materials (Mater. Today, 2019, 30: 34-51; Chem. Rev., 2021, 12: 5209-5232), and it is mainly dominated by electron transfer at the interface. For example, by utilizing the CE-driven electron transfer at the surface of PTFE, ROS could be produced even PTFE is chemically inert to a majority of chemicals and itself has rarely reported with any catalytic activity. Other representative triboelectric materials include but not limit to organics (FEP, PDMS, nylon, rubber, cellulose, etc.), in-inorganic materials (SiO_2 , Al_2O_3 , etc.), and even materials in our daily life such as copy paper (Nat. Commun., 2019, 10: 1427; Nat. Commun., 2020, 11: 2093; Adv. Funct. Mater., 2022, 2202964). Besides, owing to the CE is mainly a surface/interface effect, the CE ability of a material could also be modulated by proper surface modifications. For example, the CE ability of SiO_2 could be regulated by introducing different chemical groups on the surface (Nano Energy, 2023, 108198). In addition, the CE performance of MOF could also be enhanced by grafting different molecular groups (Nano Energy, 2023, 108433). Thus, the term of triboelectric material has enclosed a rather broad range of materials and their derivatives, which is challenging to define by several simple sentences. Usually, we will measure the CE ability to evaluate a material is suitable or not for specific applications.

Comment 2.3: line 167: Most molecules can be decomposed or degradation under the strong energy impact, it is not an except for ZrO_2 , so is the high degradation performance produced by the ZrO_2 due to the strong collision of the balls or the CE performance? Before the experimental, the author should consider whether ZrO_2 is a kind of triboelectric material?

Response 2.3: Thanks so much for your kind reminder. To evaluate the contact-electrification (CE) ability of ZrO_2 , we have utilized the exact same method as described in Supplementary Fig. 1, except the pristine polymer is replaced by ZrO_2 film (Fig. R1a). The quantity of transferred charges between ZrO_2 and ultrapure water was obtained by an electrometer and exhibited in Fig. R1b. Only 0.10 nC charges were transferred between ZrO_2 and ultrapure water, which is even lower than that of PP (0.15 nC), indicating that the CE ability of ZrO_2 is almost negligible when in contact with water. Considering the density of ZrO_2 is as high as 5.68 g/cm^3 , the 12.04 % degradation of methyl orange (MO) aqueous solution in ZrO_2 group should be mainly ascribed to the strong energy impact.

In order to present a more lucid manuscript, we have supplemented the measured CE ability of ZrO_2 and revised corresponding descriptions in the main text. All these changes have been highlighted by yellow background.

Fig. R1. The CE ability of ZrO₂ when in contact with ultrapure water. a, Schematic illustration of the setup for measurement. **b,** Measured transferred charges of a ZrO₂-based SE-TENG by repeatedly immersing it into ultrapure water.

Revision 4: Following revisions have been made in the main text.

Besides, degradation by conventional ball mill setups was also investigated as shown in Supplementary Fig. 9a. ZrO₂ is supposed to provide a much higher energy impact than pristine polymers can do during grinding, while exhibiting an almost negligible CE ability in contact with water (Supplementary Fig. 9b). Around 12.04 % degradation rate was achieved in in ZrO₂ group that should be mainly ascribed to the strong energy impact during ball milling. In contrast, the degradation rate in PDMS and PTFE groups could respectively reach 32.05 % and 98.02 %, as exhibited in Fig. 2g. This result not only implies that defects or extreme condition brought by high energy impact is not the primary reason for degradation in PDMS and PTFE groups, but also indicates the utilization of triboelectric materials and it-derived CEC may offer an efficient strategy for ball milling under mild conditions.

Revision 5: Following revisions have been made in the Supplementary Fig. 9.

Supplementary Fig. 9 | Investigations on degradation by ZrO₂ and its CE ability. a, UV-Vis spectra of MO aqueous solution before and after grinding in ZrO₂ group for 120 mins. b, Measured transferred charges of a ZrO₂-based SE-TENG by repeatedly immersing it into ultrapure water.

Comment 2.4: Author declaredly that “PTFE exhibits the highest CE performance and show the supreme performance, followed by PDMS, the PP group can hardly be electrified, thus produced a negligible amount of ROS”. It is not strict, whether the H₂O or O₂ molecular can be reduced largely depend on the LUMO orbit of the material, so the reduction ability of PTFE, PDMS and PP should be considered.

Response 2.4: Thanks so much for your professional advice. We have measured the CE ability of PTFE, PDMS, and PP, along with their performance in producing ROS, as exhibited in Supplementary Fig. 2 and Fig. 1c. These experimental results suggest that there exists a positive correlation between the CE abilities and ROS production rates of these polymers. In light of our previous research on studying CE at liquid-solid interfaces, we speculate the CE ability of these polymers should also present a certain relationship with their LUMO levels. For example, an electron will be transferred from H₂O molecules to PTFE chains upon their contact, and the LUMO orbit of PTFE should be responsible for accepting the transferred electron. Thus, we

fully agree with you on the perspective that the LUMO orbit of these polymers should play a dominant role in CE and subsequent redox reactions. In other words, the difference in LUMO orbit (or HOMO orbit when losing electrons) might be one of the underlying mechanisms for illustrating the discrepancy in CE abilities of distinct polymers. This is a rather interesting assumption that deserves detailed and comprehensive investigations in our future research on the mechanism of CE. We sincerely appreciate your insightful comments.

Comment 2.5: line 206. By using PTFE balls with different sizes, the contribution from impact energy was investigated. The result shows that the higher size radius exhibits a faster degradation due to the higher impact energy, however, with increase on balls size leads to a decay in degradation rate, and the author explained that a drastic decrease of CE frequency. So, what is the main contribution on a high degradation performance, a higher CE frequency or the stronger impact energy?

Response 2.5: Thanks so much for your kind remind. Increase in either the impact frequency or energy could significantly improve the degradation rate. However, there exists a trade-off between impact energy and frequency in the investigation on the relationship between ball size and degradation rate due to the total weight of PTFE balls is fixed at 100 g. This experimental detail has been specified in the Supplementary Note S3 but omitted in the main text. We sincerely apologize for the misunderstanding because of the unclear main text. To be specific, owing to the total weight is given, the size of milling balls would simultaneously affect the weight of individual ball and the total number of balls. The impact energy (depend on the weight of individual ball) would increase, while the impact frequency (relevant to the number of balls) would decrease along with the increase of ball sizes. Based on the qualitative model for variation tendency of ball motions in Supplementary Note 2, calculated evolutions of impact energy and frequency when the size of PTFE balls increases from 4 to 8 mm were obtained and exhibited in Fig. R2. The revolution speed in calculation is fixed at 350 RPM.

Although an apparently higher collision frequency can be obtained by employing smaller PTFE balls, the impact energy brought by these balls might be insufficient to drive interfacial electron transfer for catalysis. Thus, we expect the impact energy is the limiting factor for degradation at this stage, and the degradation rate would first rise with the increase of ball's size. This is also in accordance with the experimental observations that the degradation rate in 5-mm group is higher than in 4-mm group. However, further increases in ball sizes would bring about a significant decrease in the impact frequency, which is also a key factor that affects the CE-driven electron transfer process. On this condition, the impact energy is already beyond the threshold for driving electron transfer during CE, and the impact frequency is supposed to appear as the limiting factor for degradation. Thus, the degradation rate is expected to decrease with further increase of ball sizes, which is responsible for the result that the degradation rate in 8-mm group is lower than that in 5-mm group.

To wrap up, the evolution of degradation rate could be ascribed to that the impact energy and frequency respectively appears as the limiting factor when the ball size varies, and a trade-off relationship exists between these two factors. We have carefully revised the corresponding descriptions in the main text as well as supplementary information for presenting a clearer manuscript. Thanks again for your valuable advice!

Fig. R2. Calculated evolution of impact energy and frequency when the diameter of PTFE balls varies from 4 to 8 mm. Note that the total weight of PTFE balls is given as 100 g.

Revision 6: Following revisions have been made in the main text.

The total weight of PTFE balls was given as 100 g. The 5-mm group exhibits a faster degradation than 4-mm group, which might be ascribed to the higher impact energy brought by 5-mm group. Although an even higher impact energy could be obtained by further increasing the size of balls to 8 mm, the measured degradation rate decays significantly, which might result from a drastic decrease of CE frequency. The outperformance of 5-mm group indicates there exists as an optimal diameter to reach a balance between impact frequency and energy, as specified in Note S3.

Revision 7: Following revisions have been made in the Supplementary Information.

Supplementary Note 3: Owing to the total weight of milling balls is given as 100 g in this study, the size of milling balls would simultaneously affect the weight of the individual ball and the total number of balls. Thus, the impact energy (depend on the weight of individual ball) would increase, while the impact frequency (relevant to the number of balls) would decrease along with the increase of ball sizes. Based on the qualitative model for variation tendency of ball motions in Supplementary Note 2, calculated evolutions of impact energy and frequency when the size of PTFE balls increases from 4 to 8 mm were obtained and exhibited in Supplementary Fig. 15. The revolution speed in calculation is fixed at 350 RPM.

Although an apparently higher collision frequency can be obtained by employing smaller PTFE balls, the impact energy brought by these balls might be insufficient to drive interfacial electron transfer for catalysis. Thus, we expect the impact energy is the limiting factor for degradation at this stage, and the degradation rate would first rise with the increase of ball's size. This is also in accordance with the experimental observations that the degradation rate in 5-mm group is higher than in 4-mm group. However, further increases in ball sizes would bring about a significant decrease in the impact frequency, which is also a key factor that affects the CE-driven electron transfer process. On this condition, the impact energy should already beyond the requirement for driving electron transfer during CE, and the impact frequency is supposed to appear as the limiting factor for degradation. Thus, the degradation rate is

expected to decrease with further increase of ball sizes, which is responsible for the result that the degradation rate in 8-mm group is lower than that in 5-mm group.

To wrap up, the evolution of degradation rate could be ascribed to that the impact energy and frequency respectively appears as the limiting factor in different conditions, and a trade-off relationship exists between these two factors. Thus, there exists an optimum ball size for achieving the highest CEC efficiency by a balancing collision frequencies and impact energies.

Supplementary Fig. 15. Calculated evolution of impact energy and frequency when the diameter of PTFE balls varies from 4 to 8 mm. Note that the total weight of PTFE balls is given as 100 g.

Comment 2.6: The correlation ship among the generation of ROS, the impact energy and the CEC performance need to deep considered. And further, Fig 5a, to make more clearly, the specific value of impact energy should be given.

Response 2.6: Thanks so much for your suggestive advice. The impact energy would affect the electron transfer process during CE, which could thus modulate the subsequent redox reactions for producing ROS. Specifically, on the one hand, a higher impact energy is supposed to provide more energies for electrons to transit during CE. On the other hand, the higher impact is supposed to result in a closer contact at the interface, which is also beneficial for CE-driven interfacial electron transfer by decreasing the energy barrier (Supplementary Fig. 12d). As a consequence, the electron exchange process during CE could be greatly facilitated by the increase of impact energy, and the formation of ROS is exactly result from such CE-driven electron transfer. Therefore, an increase in the impact energy could significantly improve the production of ROS through an enhanced CE process.

As for the Fig. 5a, we admit that it would be better for comprehension if the specific value of impact energy could present together with the figure. However, we are inclined to present a general schematic diagram instead of a specific example in Fig. 5a for illustrating the existence of speed threshold for initiating CEC. Thus, we suppose it would be better to list the corresponding impact energy separately. According to the qualitative model established in Supplementary Note 2, we have calculated the specific value of impact energy under various revolution speeds and listed them in Supplementary Table 1. We sincerely hope you could understand and support.

Revision 8: Following revisions have been made in the main text.

According to the qualitative model established in Supplementary Note 2, the specific value of impact energy of 5-mm PTFE balls under various revolution speeds were calculated and listed in Supplementary Table 1.

Revision 9: Following revisions have been made in the Supplementary Information.

Supplementary Table 1. Calculated specific value of impact energy of 5-mm PTFE balls under various revolution speeds.

Revolution Speeds (RPM)	Impact energy (J)
50	0.08805
100	0.3522
150	0.79245
250	2.20126
350	4.31446

Reviewer #3 (Remarks to the Author):

Comments: The results are noteworthy, in my opinion. It does add to the existing literature in the field concerning contact electrification. I also find the methodology to be sound, but I suggest that the presentation could be improved by discussing some additional references that I think are relevant to this topic. The spontaneous electrification of surfaces and interfaces is a widespread phenomenon that produces unexpected effects in mass and charge transfer, phase change, and chemical reactivity, as revealed in abundant literature from the past twenty years. I was disappointed to find no references to the Brazilian scientists Fernando Galembeck and Andre Galembeck. I also urge that consideration be given to the recent publication: Colussi, A. J., Mechanism of Hydrogen Peroxide Formation on Sprayed Water Microdroplets. *J. Am. Chem. Soc.*, 2023, 145, 16315–16317 which I think is relevant to how contact electrification works.

Response: We express our heartfelt gratitude to the reviewer for offering this valuable and professional suggestion, which is of indispensable significance for us to improve the literature survey and scientific rigor of this manuscript. Indeed, the CE effect could provide an electric field in space, and a high-intensity electric field at the contact interfaces is able to significantly affect surrounding environments, such as facilitating the electron transfer between different substrates for catalysis. We feel sorry for omitting the discussion of impact and contribution from the electric field at contact interfaces in our previous manuscript. We have carefully revised the manuscript after reading these recommended publications in addition with some other related papers. Corresponding revisions have been made in the main text and highlighted by yellow background. Thanks again for your precious advice.

Revision 10: Following revisions have been made in the Introduction.

The CE effect is ubiquitously existed among various interfaces, and recent studies have proved that electrons are the dominant charge carriers of CE in a majority of cases.²²⁻²⁴ Besides,

a high-intensity electric field at the contact surface could facilitate the electron transfer between different substrates as well as to motivate the subsequent reactions.²⁵⁻²⁷ Contact-electro-catalysis (CEC) has been proposed to describe a catalytic process that is promoted by CE-driven interfacial electron-transfer.²⁸⁻³⁰

Revision 11: Following revision have been made in the main text.

Recent studies have revealed that electrons participate in, and in most cases, dominant the charge transfer mechanism of CE.^{23,24} Besides, the charged surface because of CE would result in an electric field in space, and a high-intensity electric field at the contact interface could significantly affect ambient environments.^{25,26} For example, the high-intensity electric field at the surface of water microdroplet is supposed to promote electron transfer for the formation of hydroxyl radicals and H₂O₂.⁴²⁻⁴⁴ Such electric field has also been found at the water-oil contact interfaces, and it-driven electron transfer could catalyze the formation of reactive oxygen species (ROS) for effective degradation of hexadecane.²⁷ A higher reaction rate could be obtained by intensify the electric field at the surface.⁴⁴ These experimental observations further support the feasibility of catalyzing chemical reactions by CE.

REVIEWERS' COMMENTS

Reviewer #2 (Remarks to the Author):

The authors have addressed my concerns raised before and made revisions properly. The present work is now acceptable for publication.

Response to Referees' Comments

Reviewer #2 (Remarks to the Author):

Comments:

The authors have addressed my concerns raised before and made revisions properly. The present work is now acceptable for publication.

Response: We are very grateful for your positive feedback on the response and revised manuscript. Your thoughtful comments and suggestions have undoubtedly contributed to the improvement of our work. Thank you once again for your time and expertise.